# Impact of Preoperative Yttrium-90 Transarterial Radioembolization on Patients Undergoing Right or Extended Right Hepatectomy for Hepatocellular Carcinoma

**DOI:** 10.3390/cancers17152556

**Published:** 2025-08-02

**Authors:** Andrea P. Fontana, Nadia Russolillo, Ludovica Maurino, Andrea Marengo, Amedeo Calvo, Andrea Ricotti, Serena Langella, Roberto Lo Tesoriere, Alessandro Ferrero

**Affiliations:** 1Department of General and Oncological Surgery, Mauriziano Hospital, 10128 Torino, Italy; nrussolillo@mauriziano.it (N.R.); ludovica.maurino@studio.unibo.it (L.M.); slangella@mauriziano.it (S.L.); rlotesoriere@mauriziano.it (R.L.T.); aferrero@mauriziano.it (A.F.); 2Department of Gastroenterology, Mauriziano Hospital, 10128 Torino, Italy; amarengo@mauriziano.it; 3Department of Interventional Radiology, Mauriziano Hospital, 10128 Torino, Italy; acalvo@mauriziano.it; 4Clinical Trial Unit, AO Ordine Mauriziano Hospital, 10128 Torino, Italy; aricotti@mauriziano.it

**Keywords:** hepatocellular carcinoma (HCC), Yttrium-90 transarterial radioembolization (TARE), major hepatectomy, neoadjuvant therapy, macrovascular invasion

## Abstract

Hepatocellular carcinoma (HCC) is often diagnosed at an advanced stage, limiting curative-intent treatment options. For patients with large tumors or those showing macrovascular invasion, major hepatectomy may be required, but carries substantial risk. This study investigated the impact of Yttrium-90 transarterial radioembolization (TARE) before surgery in patients undergoing right or extended right hepatectomy for HCC. Among 39 patients, 18 received preoperative TARE and 21 underwent upfront resection. While perioperative outcomes were similar, the TARE group demonstrated significantly better long-term oncological results, including higher cancer-specific survival, recurrence-free survival, and curative-intent disease-free survival. TARE also led to greater tumor necrosis and enabled more extended resections without increasing surgical risk. These findings suggest that TARE may be a safe and effective neoadjuvant option in selected patients with borderline resectable HCC, including those with macrovascular invasion, potentially improving outcomes and aiding surgical decision-making.

## 1. Introduction

Hepatocellular carcinoma (HCC) is the most common form of liver cancer, accounting for approximately 90% of all primary liver cancer cases [1]. With over 900,000 new cases annually and 830,000 HCC-related deaths recorded, it represents the sixth most common cancer and the third most common reason for cancer-associated death in the world [2,3]. To date, approximately half of patients diagnosed with hepatocellular carcinoma present with disease stages that are not amenable to curative treatment, according to the most widely accepted international guidelines [1,4,5,6,7,8].

However, profound changes are currently reshaping the approach to HCC management.

The etiology of chronic liver disease is shifting from viral hepatitis-related cirrhosis to Metabolic dysfunction-Associated Liver Disease (MAFLD), driven on one hand by the success of HBV vaccination campaigns, the effectiveness of antiviral therapies for HCV, and improved screening programs, and, on the other hand, by the rising prevalence of obesity and type 2 diabetes [9,10].

The recent approval of truly effective systemic therapies [11,12], together with increased safety of surgical treatments [13,14,15,16], has introduced a degree of flexibility into treatment paradigms that have traditionally been highly rigid [6]. Emerging concepts such as treatment stage migration, neoadjuvant therapy, and conversion strategies are beginning to gain recognition in the management of HCC, as has already occurred in most other solid malignancies [5,6,17]. In this evolving therapeutic scenario, the role of neoadjuvant approaches aimed at expanding surgical indications is gaining increasing attention. Yttrium-90 transarterial radioembolization (TARE) is a well-established locoregional therapy in the multimodal management of HCC, with robust evidence supporting its efficacy in achieving tumor control and improving progression-free survival in selected patients [5,18,19,20]. However, TARE is currently classified among palliative treatments in most international guidelines, and its role as a conversion strategy remains controversial. The latest update of the Barcelona Clinic Liver Cancer (BCLC) staging system as well as the most recent version of the European Association for the Study of the Liver clinical practice guidelines has incorporated TARE within its treatment algorithm, but only for single lesions <8 cm, based on the results of the LEGACY and XXL studies [5,6,21,22].

Nevertheless, TARE holds significant potential as a neoadjuvant treatment. Its ability to induce not only tumor necrosis but also ipsilateral parenchymal atrophy and contralateral hypertrophy of the future liver remnant (FLR) makes it an attractive option in patients with marginally resectable tumors requiring major hepatectomy. Moreover, unlike other locoregional therapies such as transarterial chemoembolization (TACE) or hypertrophy-inducing strategies like portal vein embolization (PVE), TARE can be safely employed even in patients with portal vein tumor thrombosis (PVTT), traditionally considered a contraindication to liver resection.

Surgical treatments, namely liver resection and liver transplantation, alongside thermal ablation, currently represent the only potentially curative options for patients with hepatocellular carcinoma [5,6,17]. While transplantation offers the advantage of treating both the tumor and the underlying liver disease, liver resection remains a cornerstone in the management of patients with preserved liver function and tumors not meeting transplant criteria. In this setting, anatomical but parenchymal-sparing liver resections (PSLRs) are recommended to minimize the risk of postoperative liver failure, especially in cirrhotic patients [23,24,25].

However, PSLR is not always feasible, particularly in cases of large tumors (>5 cm) and/or macrovascular invasion, where a major or even extended hepatectomy may be necessary. In such scenarios, accurate assessment of the future liver remnant (FLR), both in terms of volume and function, is crucial to reduce the risk of post-hepatectomy liver failure (PHLF). When the FLR is deemed insufficient, preoperative strategies to induce hypertrophy, such as portal vein embolization (PVE) or liver venous deprivation (LVD) or, as already mentioned, TARE, can be employed to improve surgical candidacy.

The aim of the present study was to investigate the short- and long-term outcomes of a group of patients, all undergoing right and right extended hepatectomy for hepatocellular carcinoma with or without prior Yttrium-90 transarterial radioembolization (TARE).

## 2. Materials and Methods

### 2.1. Patient Selection

Data from all consecutive patients who underwent curative-intent liver resection for hepatocellular carcinoma between January 2013 and December 2023 at the Department of General and Oncological Surgery, Mauriziano Hospital, Turin, Italy, were retrospectively extracted from a prospectively maintained institutional database. Patients were included starting from January 2013, as this marked the introduction of TARE at our institution. More recent cases were excluded to ensure a minimum follow-up period of 18 months for all patients.

Inclusion criteria were age over 18 years, right or extended right hepatectomy performed, and availability of follow-up data. Exclusion criteria included repeat liver resections, two-stage hepatectomy, R2 resections, presence of extrahepatic disease, or a final pathological diagnosis other than hepatocellular carcinoma.

The primary endpoint of this study was to compare long-term oncological outcomes between patients who underwent TARE prior to surgery and those who underwent upfront resection. Outcomes evaluated included overall survival (OS), cancer-specific survival (CSS), recurrence-free survival (RFS), and curative-intent disease-free survival (ciDFS). Secondary endpoints included postoperative outcomes and histopathological response to TARE.

This study was conducted in accordance with the principles of the Declaration of Helsinki and its amendments. Approval for data collection and analysis was obtained from the Institutional Review Board of Città della Salute e della Scienza University Hospital, Turin, Italy. The ethics approval code is 00512/2020, and the approval was granted on 14 December 2020.

### 2.2. Diagnosis and Preoperative Staging

The diagnosis of HCC was established based on radiological criteria according to international guidelines, with percutaneous biopsy performed only in cases with inconclusive imaging findings [5,9]. All patients underwent staging with high-resolution thoraco-abdominal computed tomography (CT), complete liver function tests, alpha-fetoprotein (AFP) measurement, and esophagogastroduodenoscopy (EGDS). Although no functional imaging was used to assess future liver remnant (FLR) function, the indocyanine green retention rate at 15 min (15′ ICG-RR) was measured in all patients, following the Makuuchi protocol for hepatic functional reserve assessment [26].

In cases of uncertain diagnosis, suspected multifocal disease, or biliary involvement, magnetic resonance imaging (MRI) with a liver-specific contrast agent (gadoxetic acid, Gd-EOB-DTPA; Primovist^®^ Bayer AG, Berlin, Germany) combined with magnetic resonance cholangiopancreatography (MRCP) was also performed.

Preoperative macrovascular invasion of the portal and hepatic veins was assessed using CT according to the tumor thrombosis classification of the Liver Cancer Study Group of Japan [27,28].

Indications for liver resection were discussed in a multidisciplinary team (MDT) meeting, following a therapeutic hierarchy rather than a stage-based hierarchy [17]. Candidates for surgery were required to have good overall health status, defined as an Eastern Cooperative Oncology Group (ECOG) performance status of 0 or 1, and preserved liver function, defined as a Child Pugh score ≤ B7. Patients with signs of portal hypertension, based on laboratory findings (platelet count < 100,000/μL), endoscopy (presence of esophageal varices), and/or CT imaging (portosystemic collaterals, splenomegaly), were evaluated on a case-by-case basis. Endoscopic band ligation of esophageal varices was performed in all patients prior to undergoing liver resection. Comorbidity assessment was performed using both the Charlson Comorbidity Index [29] and the American Society of Anesthesiologists (ASA) physical status classification system [30].

An adequate functional reserve of the FLR was also mandatory. The ratio between the future liver remnant and the total liver volume, excluding the tumoral volume (FLR/TLV ratio), had to be >30% for non-cirrhotic livers and >40% for cirrhotic ones. In patients scheduled for upfront surgery, liver volumetry was systematically performed. In cases with inadequate volume (FLR/TLV < 30%), portal vein embolization (PVE) was performed to induce compensatory hypertrophy when TARE was not indicated by the multidisciplinary team (MDT). In patients undergoing preoperative TARE, both pre- and post-procedural volumetry were conducted to assess the actual hypertrophy achieved.

### 2.3. TARE Protocol and Technique

All patients included in the present study who underwent TARE were treated using 90Y-resin microspheres (SIR-Spheres^®^, Sirtex Medical Europe GmbH), according to a standardized institutional protocol established at Ordine Mauriziano Hospital, Turin and previously described [31,32]. Pre-treatment angiographic evaluation was systematically performed to delineate hepatic arterial anatomy and identify any extrahepatic arterial branches. Embolization of vessels such as the right gastric and gastroduodenal arteries was carried out when necessary to prevent non-target deposition of microspheres.

A simulation procedure followed, consisting of intra-arterial administration of 150 MBq of technetium-99m-labeled macroaggregated albumin (^99mTc-MAA, LyoMAA^®^, Mallinckrodt). Whole-body planar scintigraphy and single-photon emission computed tomography (SPECT)/CT were used to evaluate pulmonary shunting and rule out gastrointestinal or spinal cord extrahepatic uptake. Patients with significant extrahepatic shunting were excluded from treatment.

Dosimetry was calculated using both the partition model (Medical Internal Radiation Dose, MIRD) and a voxel-based convolution method developed in-house with MATLAB^®^ (version R2020b, MathWorks Inc., Natick, MA, USA), based on ^99mTc-MAA distribution. The objective was to achieve a tumor-absorbed dose >120 Gy, while keeping the dose to normal liver parenchyma and lungs below 40 Gy and 30 Gy, respectively. Final administered activity ranged between 1.5 and 2.2 GBq, tailored to each patient’s anatomy and tumor burden.

Radioembolization was performed within two weeks of the simulation. A microcatheter was advanced into the right hepatic artery and ^90Y-resin microspheres were infused under fluoroscopic guidance. No embolic agents other than the microspheres were employed. Post-treatment verification of microsphere distribution and dose delivery was performed with a SPECT/CT acquired within 24 h.

Patients were monitored after the procedure with liver function tests and CT scan at 3 months to assess radiologic response, and to measure FLR hypertrophy. A second TARE session was considered in patients with suboptimal biological or volumetric response. All procedures were performed in coordination with interventional radiology, nuclear medicine, medical physics, and surgical teams.

### 2.4. Surgical Technique

Surgical procedures were performed according to the standard technique adopted at our institution, as previously described [33,34,35]. All surgeries were conducted using either an open or laparoscopic approach, as the robotic-assisted technique was not yet implemented during the study period.

Patients were positioned supine with a slight reverse Trendelenburg tilt. For the open approach, the legs were kept together; for the laparoscopic approach, a split-leg position was adopted to allow the surgeon to operate between the patient’s legs. In the open approach, a right subcostal incision extended 7 cm to the left was performed. For the laparoscopic approach, pneumoperitoneum was established through a 12 mm right pararectal trocar, and a 30° laparoscope was used. Four 12 mm trocars and one 5 mm trocar were placed along a curved line from the xiphoid process to the right costal margin. In cases of conversion to open surgery, the trocar incisions were joined to create a J-shaped right subcostal incision.

After abdominal exploration, intraoperative ultrasonography (IOUS) was performed to assess the extent of disease, using the BK Medical Pro Focus 2202 system. The hepatic hilum was controlled with a nylon tape and tourniquet and cholecystectomy was subsequently performed.

An extrahepatic intra-Glissonian approach was used to isolate and divide the right hepatic artery and right portal vein, only after visual and IOUS confirmation of left hepatic artery and portal vein patency. The right hemiliver was fully mobilized up to the retrohepatic inferior vena cava (IVC). The hepatocaval ligament and any accessory right hepatic veins were ligated and divided. In cases of right hepatectomy extended to segment 1, mobilization of segment 1 was also performed at this stage.

The right hepatic vein (RHV) was usually dissected extrahepatically and controlled with a vessel loop, or divided at this stage when mobilization of segment 1 was required. Ischemic demarcation of the right liver was outlined using electrocautery, typically extending to the confluence of the middle hepatic vein (MHV) with the inferior vena cava (IVC). In laparoscopic cases, indocyanine green (ICG)-guided resection was routinely performed using a negative-staining technique.

Parenchymal transection was carried out using an ultrasonic dissector (SonaStar Laparoscopic Probe MXA-L002) and a radiofrequency sealer/divider (LigaSure^TM^, Covidien). The RHV was divided using an endovascular stapler. The right hepatic duct was transected within the liver parenchyma.

For very large tumors (>10 cm) involving the right posterior sector, an anterior approach was adopted. In such cases, due to tumor size, an open approach was preferred. The space between the MHV and RHV was dissected cranially using an aortic clamp, and the liver was suspended with surgical tape. IOUS was used to guide retrohepatic dissection during the hanging maneuver. Parenchymal transection was carried out from the anterior surface toward the IVC without prior mobilization of the right liver. The RHV was divided at the end of the resection using an endovascular stapler, after which liver mobilization was completed.

Regardless of the surgical approach or technique, an intracorporeal Pringle maneuver was applied when necessary.

Reconstruction of the falciform ligament was routinely performed at the end of the procedure. Surgical drains were placed based on intraoperative assessment, particularly in cases of significant blood loss or risk of biliary leakage.

### 2.5. Intra- and Postoperative Outcomes

Intraoperative blood loss was measured by summing the fluids collected in the suction containers and subtracting the volume of intraoperative irrigation fluids used. Right hepatectomy and extended right hepatectomy involving segment 4 and/or segment 1 were defined according to the Brisbane 2000 terminology [36]. Intraoperative complications were defined as iatrogenic injuries or other significant adverse events occurring during surgery that resulted in a deviation from the planned intraoperative course (e.g., major bleeding, air embolism, unexpected vascular or biliary repairs).

Postoperative complications were documented and classified according to the Clavien–Dindo (CD) system, with severe complications defined as CD grade III or higher [37]. Major complications were also defined by a Comprehensive Complication Index (CCI) greater than 20.9 [38]. Bile leakage and post-hepatectomy liver failure (PHLF) [39,40] were classified according to the definitions provided by the International Study Group of Liver Surgery (ISGLS). Postoperative mortality was defined as any death occurring during the same hospital admission or within 90 days after liver resection.

Histological examination of the surgical specimen allowed for the assessment of the quality of the non-tumoral liver parenchyma (e.g., cirrhosis, steatosis, fibrosis, normal tissue), the degree of HCC differentiation according to Edmondson-Steiner grading [32], the number of nodules, the presence of a capsule and satellite nodules, microvascular and macrovascular invasion, and the presence and extent (percentage) of tumor necrosis.

### 2.6. Follow-Up

After surgery, patients were followed-up with abdominal ultrasound and blood tests—including complete blood count, liver and renal function tests, and AFP—every 3 months for the first 2 years, every 4 months for the subsequent 3 years, and every 6 months thereafter, up to a maximum follow-up of 10 years. A thoraco-abdominal CT scan was performed annually.

In cases of suspected or confirmed recurrence, patients were re-evaluated during an MDT meeting, following the same therapeutic hierarchy applied at the time of initial diagnosis. When recurrence was confined to the liver and limited in extent, potentially curative treatments such as repeat hepatectomy or thermal ablation were considered, depending on tumor location, liver function, and performance status. Systemic or locoregional therapies were discussed for cases not amenable to curative treatment.

### 2.7. Definitions of Long-Term Outcomes

Patients who died within 90 days after surgery were excluded from the long-term outcomes analysis.

Overall survival (OS) was defined as the time interval between the date of diagnosis and death from any cause.

Cancer-specific survival (CSS) was defined as the time from diagnosis to death directly attributable to the malignancy.

Recurrence-free survival (RFS) was defined as the time from surgery to the first documented recurrence, regardless of its location or the type of treatment received.

Curative-intent disease-free survival (ciDFS) was defined as the period from surgery to either death or the first recurrence not amenable to curative treatment. Patients who developed isolated intrahepatic recurrences treated with curative intent (i.e., surgical resection or thermal ablation) were not considered to have experienced an event and were still classified as disease-free under this definition.

### 2.8. Statistical Analysis

Results are presented as median values and interquartile ranges [IQR] for continuous variables, and as absolute numbers and percentages for categorical variables. Differences between groups were assessed using the Mann–Whitney U test for continuous data and the Chi-squared or Fisher’s exact test for categorical data, as appropriate.

Overall survival (OS), cancer-specific survival (CSS), and recurrence-free survival (RFS) were estimated using the Kaplan–Meier method and compared between groups using the log-rank test. Survival rates at 1, 3, and 5 years were reported along with their 95% confidence intervals (CIs).

To further assess the association between preoperative TARE and long-term oncological outcomes, a univariate Cox proportional hazards regression analysis was performed using TARE as the sole explanatory variable. Hazard ratios (HRs) with corresponding 95% CIs were reported to estimate the risk of events in the TARE versus non-TARE group.

All statistical tests were two-sided, and a *p*-value < 0.05 was considered statistically significant. Analyses were performed using R software version 4.3.3 (R Core Team, 2024) [41].

## 3. Results

Overall, 249 patients affected by HCC who underwent liver resection with curative intent between January 2013 and December 2023 were identified. A total of 39 patients were included in the final analysis. The reasons for case exclusion are illustrated in the flow diagram shown in Figure 1.

Of the 39 patients who underwent right or right extended hepatectomy, 18 received preoperative TARE, whereas 21 did not. The median tumor-absorbed dose in the TARE group was 230.5 Gy [139.5–351.0], as calculated using patient-specific voxel-based dosimetry. In 2 cases, TARE was indicated but not performed due to the presence of extrahepatic shunts (to the lung in one case and to the spinal cord in the other); both patients underwent upfront surgery. In 4 patients from the non-TARE group, TACE followed by PVE was performed. None of the patients in the TARE group required PVE. The median interval between TARE and surgery was 21 [15,16,17,18,19,20,21,22,23,24,25,26,27,28,29,30,31,32,33] weeks.

### 3.1. Baseline Characteristics

Patient and tumor characteristics at baseline are summarized in Table 1. Overall, baseline characteristics were comparable between the two groups.

However, two significant differences were observed. The TARE group had a higher prevalence of virus-related liver disease, with 7 patients (38.9%) vs. 6 (28.6%) presenting HCV-related hepatopathy, and 4 patients (22.2%) vs. 0 with HBV-related hepatopathy (*p* = 0.033). Additionally, the TARE group showed significantly higher 15 min indocyanine green retention rate (ICG-R15) values: 14.80 [9.85–21.73] vs. 10.30 [4.30–12.50] (*p* = 0.033).

Comorbidity burden was comparable between the two groups, as reflected by similar Charlson Comorbidity Index scores and ASA classifications. Most patients in both groups were classified as Child Pugh class A (scores 5–6), while two patients in the non-TARE group were classified as Child Pugh B7. No significant differences were observed between the groups regarding preoperative bilirubin levels.

The majority of patients presented with a large single lesion, with median tumor sizes of 70 mm in the TARE group and 80 mm in the non-TARE group (*p* = 0.171). Multifocal disease was observed in 8 patients (44.4%) in the TARE group and in 3 patients (14.3%) in the non-TARE group. However, the differences in the number of nodules and tumor size were not statistically significant.

Radiologic evidence of macrovascular invasion was identified in 23 patients (59%) across the cohort. Specifically, portal vein invasion was observed in 7 patients (17.9%), hepatic vein invasion in 10 (25.6%), and concomitant portal and hepatic vein invasion in 6 (15.4%). The distribution of vascular involvement is detailed in Table 1. A statistically significant difference between the two groups was observed for portal vein invasion/thrombosis, particularly in patients classified as VP3, defined as invasion or thrombosis of the right portal vein. No significant differences were found regarding hepatic vein invasion, concomitant portal and hepatic vein invasion or overall macrovascular invasion.

BCLC staging was retrospectively assigned for all patients based on clinical and radiological data at diagnosis. No significant differences in BCLC stage distribution were observed between the TARE and non-TARE groups. Although 23 patients had macrovascular invasion, only 20 were classified as BCLC stage C. This discrepancy is due to the BCLC staging definition, which considers patients as stage C only in cases of portal vein involvement up to second-order branches (Vp2–Vp3) or main hepatic vein/inferior vena cava invasion (VV2–VV3). In contrast, three patients had macrovascular invasion limited to third-order portal vein branches (Vp1) or minor hepatic vein branches (VV1), and were not assigned to stage C accordingly.

The median future liver remnant (FLR) in the TARE group increased from 692.50 cm^3^ [485.50–769.47], corresponding to a FLR/TLV ratio of 39.1% [35.75–50.72] pre-treatment, to 767.14 cm^3^ [697.18–872.75] post-treatment, with a corresponding FLR/TLV ratio of 56.9% [40.00–62.25].

In the non-TARE group, the median FLR was 574.20 cm^3^ [510.66–748.18], corresponding to a FLR/TLV ratio of 40.0% [37.0–47.3] (*p* = 0.949).

Among the four patients in the non-TARE group who underwent TACE followed by PVE, the median FLR after PVE was 664.70 cm^3^ [648.60–741.85], with a corresponding FLR/TLV ratio of 51.9% [51.0–53.9] (*p* = 0.737).

### 3.2. Intraoperative Outcomes

Intraoperative characteristics are summarized in Table 2. Overall, surgical outcomes were comparable between the two groups.

However, extended right hepatectomies were significantly more frequent in the TARE group compared to the non-TARE group (55.6% vs. 19.0%, *p* = 0.024).

A laparoscopic approach was adopted in five patients (23.8%) in the non-TARE group and in eight patients (44.4%) in the TARE group (*p* = 0.196). One planned, non-emergent conversion to open surgery occurred in the TARE group due to diaphragmatic infiltration, which required partial diaphragmatic resection.

### 3.3. Postoperative Outcomes and Histological Findings

Postoperative characteristics are summarized in Table 3. Overall, postoperative outcomes were comparable between the two groups.

No significant differences were observed in terms of length of hospital stay, minor or major postoperative complications, need for blood transfusions, 90-day mortality, or hospital readmission rates. Three patients (7.7%) in the entire cohort died within 90 days after surgery: one in the TARE group (5.6%) and two in the non-TARE group (9.5%), respectively (*p* = 1). All deaths were due to sepsis and multiorgan failure in the context of post-hepatectomy liver failure.

Histopathological analysis revealed a significantly larger tumor size in the non-TARE group compared to the TARE group (80.0 mm vs. 46.5 mm, *p* = 0.020), as well as a markedly higher percentage of tumor necrosis in the TARE group (70% vs. 10%, *p* = 0.002). Additionally, resection margins were significantly wider in the non-TARE group (10 mm vs. 2 mm, *p* = 0.032).

### 3.4. Long-Term Outcomes

The three patients (7.7%) from the entire cohort who died within 90 days of surgery were excluded from the long-term outcomes’ analysis.

Median follow-up duration was 39.5 months in the TARE group and 31.4 months in the non-TARE group (*p* = 0.428). A total of 6 patients (35.3%) in the TARE group and 12 patients (63.2%) in the non-TARE group died during the study period. Among the TARE group, three deaths were due to causes unrelated to HCC: two were attributable to cardiovascular disease and one to gastric cancer diagnosed after HCC surgery. All remaining deaths in both groups were related to HCC progression.

Tumor recurrence occurred in 10 patients (58.8%) in the TARE group and 16 patients (84.2%) in the non-TARE group. In the TARE group, all recurrences were intrahepatic. Of these, five patients (50.0%) presented with a solitary liver recurrence and were treated with curative intent using radiofrequency ablation (RFA) or microwave ablation (MWA). One patient received transarterial chemoembolization (TACE), three received systemic therapy, and one patient (10%) was managed with best supportive care (BSC).

In the non-TARE group, 11 patients (68.8%) had liver-only recurrence. Among these, one patient (6.3%) had a solitary nodule treated with RFA with curative intent, two patients underwent TACE followed by sorafenib, three received systemic therapy alone, and five (31.3%) were managed with BSC. Two additional patients developed extrahepatic recurrence and were treated with systemic therapy, while three patients experienced both hepatic and extrahepatic recurrence and also received systemic treatment.

A comparison of the Kaplan–Meier curves (Figure 2) showed a higher overall survival (OS) in the TARE group, although this difference did not reach statistical significance (*p* = 0.1). The 1-, 3-, and 5-year OS rates were 94.1% (95% CI: 0.84–1.00), 80.2% (95% CI: 0.62–1.00), and 68.8% (95% CI: 0.46–1.00) in the TARE group, and 84.2% (95% CI: 0.69–1.00), 50.2% (95% CI: 0.31–0.80), and 33.5% (95% CI: 0.00–0.70) in the non-TARE group, respectively.

When considering cancer-specific survival (CSS), the Kaplan–Meier curves’ comparison revealed a statistically significant difference in favor of the TARE group (*p* = 0.011). The 1-, 3-, and 5-year CSS rates were 100% (95% CI: 1.00–1.00), 93.8% (95% CI: 0.82–1.00), and 80.4% (95% CI: 0.58–1.00) in the TARE group, and 84.2% (95% CI: 0.69–1.00), 50.2% (95% CI: 0.31–0.80), and 33.5% (95% CI: 0.16–0.69) in the non-TARE group, respectively.

Furthermore, the Kaplan–Meier curves showed a significantly higher recurrence-free survival (RFS) in the TARE group compared to the non-TARE group (*p* = 0.047). The 1-, 3-, and 5-year RFS rates were 82.4% (95% CI: 0.66–1.00), 42.2% (95% CI: 0.23–0.77), and 33.8% (95% CI: 0.16–0.71) in the TARE group, and 36.8% (95% CI: 0.20–0.66), 14.0% (95% CI: 0.043–0.45), and 14.0% (95% CI: 0.043–0.45) in the non-TARE group, respectively.

Finally, the Kaplan–Meier curves showed a significantly higher curative-intent disease-free survival (ciDFS) in the TARE group compared to the non-TARE group (*p* = 0.0037). The 1-, 3-, and 5-year ciDFS rates were 88.2% (95% CI: 0.74–1.00), 69.3% (95% CI: 0.50–1.00), and 69.3% (95% CI: 0.50–1.00) in the TARE group, and 42.1% (95% CI: 0.24–0.71), 18.9% (95% CI: 0.07–0.50), and 18.9% (95% CI: 0.07–0.50) in the non-TARE group, respectively.

The Cox analysis was consistent with these results. Preoperative TARE was identified as a protective factor for overall survival (HR = 0.45, 95% CI: 0.17–1.20, *p* = 0.112), cancer-specific survival (HR = 0.22, 95% CI: 0.06–0.78, *p* = 0.019), recurrence-free survival (HR = 0.45, 95% CI: 0.20–1.01, *p* = 0.051), and curative-intent disease-free survival (HR = 0.25, 95% CI: 0.09–0.69, *p* = 0.007).

## 4. Discussion

This study assessed the role of preoperative Yttrium-90 transarterial radioembolization in patients undergoing right or extended right hepatectomy for hepatocellular carcinoma, focusing on both perioperative safety and long-term oncological efficacy.

Although most patients in this study presented with adverse prognostic features traditionally associated with unresectable disease, such as large tumor size, multifocality, and macrovascular invasion, it is important to note that, in our center as well as in several other European and Eastern institutions, such patients would not be categorically considered unresectable. Indeed, our treatment philosophy, aligned with the multiparametric therapeutic hierarchy model [17], emphasizes individualized strategy over rigid stage-based decision-making. In this framework, resectability is determined not solely by tumor burden but through a comprehensive assessment incorporating technical feasibility, liver function, patient condition, and tumor biology. Nevertheless, all patients included in this cohort required major or extended liver resection and met the criteria for what can be defined as “borderline resectable” disease. In this challenging setting, TARE was employed as a neoadjuvant therapy to facilitate curative-intent surgery, offering both oncological control and dynamic selection of surgical candidates.

Baseline demographic and tumor characteristics were well balanced between the TARE and non-TARE groups. Although the TARE group exhibited higher rates of 15 min indocyanine green retention (ICG-R15), portal vein invasion/thrombosis (PVTT), and extended right hepatectomies—suggesting a slightly more advanced disease—there were no significant differences in intraoperative parameters or short-term postoperative outcomes, including operative time, blood loss, complication rates, and 90-day mortality. These findings suggest that TARE does not increase perioperative risk.

Importantly, the senior authors of the present study were also the surgeons who performed all procedures, and they consistently noted increased fibrosis and inflammation in TARE-treated patients, particularly in the hepatic hilum and during liver mobilization. Parenchymal transection was associated with increased fragility of the venous walls, which posed additional challenges in achieving hemostasis. Despite this increased technical complexity, no detrimental impact was observed on intraoperative outcomes or postoperative morbidity, further supporting the safety of this approach when conducted by experienced surgical teams.

A further benefit of TARE emerged from the volumetric data. In line with previous studies, the median increase in future liver remnant (FLR) volume after TARE was comparable to that observed in patients undergoing sequential TACE and PVE, highlighting the dual hypertrophic and oncologic efficacy of the technique, the so-called “radiation lobectomy” [42,43]. Importantly, the TARE group included patients with VP2 and VP3 portal vein tumor thrombosis (PVTT), according to the Japanese classification, conditions that typically contraindicate both TACE and PVE. This underscores a key advantage of TARE: its feasibility in the setting of portal vein invasion or thrombosis, which was not only well tolerated but, in some cases, even associated with radiological regression of PVTT between treatment and surgery, alongside a significant volumetric increase of the contralateral hemiliver. In our center, macrovascular invasion—although associated with poorer prognosis—is not considered an absolute contraindication to surgery. As previously discussed, we adopt a multiparametric therapeutic hierarchy model rather than a prognostic hierarchy model, allowing for individualized treatment strategies. Within this framework, TARE offers multiple advantages: it treats both the primary tumor and the vascular thrombus, promotes hypertrophy of the future liver remnant, and could act as a biological selector. Patients who respond favorably to TARE, including those with regression of PVTT, may be identified as suitable candidates for surgery, while those who progress despite treatment are spared major or extended hepatectomy. Therefore, this strategy may be particularly valuable in highly selected BCLC C patients without extrahepatic disease, for whom TARE enables not only downstaging and functional hypertrophy, but also dynamic selection based on tumor biology and treatment response.

Another notable finding was the significantly higher rate of tumor necrosis at final histopathology in the TARE group, alongside substantial tumor downsizing. These results underscore the high biological activity of the treatment. Nonetheless, necrosis and downsizing were heterogeneous. In some patients, the effects were modest, likely due to arterioportal or arteriovenous shunting that impeded optimal microsphere deposition. Indeed, two patients were excluded from TARE altogether due to non-correctable shunting and underwent upfront surgery instead. These findings highlight the critical role of achieving adequate tumor dosing, an increasingly recognized determinant of TARE efficacy, as supported by the DOSISPHERE-01 trial and other studies advocating for personalized, multi-compartment dosimetry over standard dosing regimens [19,20,44].

Nonetheless, the most relevant finding of the present study lies in the oncological outcomes. Overall survival (OS) demonstrated a trend, although not statistically significant, in favor of the TARE group, while cancer-specific survival (CSS) reached statistical significance, as did recurrence-free survival (RFS). Notably, curative-intent disease-free survival (ciDFS)—defined as the time from surgery to the first recurrence not amenable to radical treatment—was markedly better in the TARE group. This result can be explained by the pattern of recurrence, which was distinctly more favorable in patients who received TARE. All recurrences in the TARE group were confined to the liver, and half of these patients (5 cases, 50%) underwent potentially curative thermal ablation. Only one patient (10%) in this group was not eligible for any further treatment and was transitioned to best supportive care (BSC).

In contrast, while 68.8% of recurrences in the non-TARE group were also liver-only, curative treatment was achieved in just one patient (6.3%) via radiofrequency ablation, and a significantly higher proportion (31.3%) received BSC. This difference may reflect a combination of factors, including favorable biological selection—patients with early progression post-TARE were excluded from surgery—and the improved oncological efficacy of the surgical procedure itself, likely supported by the increased tumor necrosis observed in the resected specimens. TARE thus appears to provide both a biological filter and an oncologically favorable setting for surgery.

This is consistent with findings from a recent study by Tzedakis et al., in which the authors evaluated patients with large (≥5 cm) single HCCs undergoing either upfront resection or TARE followed by surgery. Notably, after propensity score matching, patients resected post-TARE had significantly improved overall survival compared with those undergoing upfront resection, while disease-free survival was similar. The authors interpreted this as a possible effect of biological selection, increased tumor necrosis, and the debulking effect of TARE, supporting its use as a neoadjuvant strategy in selected cases.

Although several studies have now evaluated TARE as a neoadjuvant or conversion therapy before liver resection, most have limitations in design or scope: they either include heterogeneous patient populations [42,45], or patients treated only as a bridge to transplant [46], or do not directly compare this with upfront surgery [31,47,48,49], or do not consider OS and DFS among primary endpoints [50,51,52]. Moreover, the role of TARE in the management of intermediate–advanced HCC has been supported by systematic reviews and meta-analyses [53], suggesting meaningful survival benefits, particularly in patients with portal vein involvement and preserved liver function. In this light, our study contributes valuable comparative data, reinforcing the clinical rationale for using TARE in the preoperative setting, particularly for borderline resectable HCC. Our findings align with the growing body of evidence and suggest that, in selected patients, TARE may act as a bridge to curative-intent surgery.

Furthermore, although several studies have demonstrated that upfront surgery can be safely and effectively performed even in patients with intermediate or advanced BCLC stages [28,54,55], the increasing availability of biologically active locoregional therapies such as TARE prompts a re-evaluation of treatment strategies, including in patients who are technically resectable. This is particularly relevant for the subset of patients with borderline resectable disease in the setting of underlying liver dysfunction, a clinical scenario where the oncological benefit of surgery must be carefully weighed against the risk of postoperative decompensation and recurrence. In this context, TARE offers not only local tumor control but also serves as a dynamic selection tool, enabling real-time assessment of tumor biology and treatment response prior to committing to major hepatectomy. This paradigm is further supported by the recent introduction of highly effective systemic therapies, including immunotherapy and targeted agents [11,56], as well as new and potentially more effective types of radioembolization such as Holmium-166-based treatments [57,58], which are reshaping the therapeutic landscape of HCC. Within this evolving framework, surgery must remain the mainstay of curative treatment, but its timing and indication should be increasingly guided by a multimodal and personalized approach. In selected cases, neoadjuvant strategies such as TARE may help to avoid futile operations by identifying early tumor progression or lack of response, thereby sparing high-risk patients from major resections with limited survival benefit, an approach already well established in the treatment of other tumors.

The present study has several limitations. Its retrospective and observational design, combined with the limited sample size and relatively long study period, warrants caution in the interpretation of the results. In addition, the decision to proceed with TARE was made during multidisciplinary tumor board meetings, and patients with similar clinical profiles occasionally received different treatment strategies. This heterogeneity complicates the definition of precise selection criteria and could have introduced a degree of selection bias. However, the study specifically addresses a selected population of patients with hepatocellular carcinoma considered borderline resectable. This subgroup is often excluded from large prospective trials due to the complexity of their clinical condition. In this context, retrospective analyses provide important real-world data and reflect actual clinical practice. The findings offer insight into how TARE can be integrated within a multimodal treatment strategy for patients who would otherwise be managed with palliative intent. Moreover, all patients were managed within a single tertiary center by a consistent multidisciplinary team. Standardized protocols for both treatment and surgery were applied. Baseline demographic and tumor characteristics were comparable across the groups. These factors strengthen the reliability and internal consistency of the findings, and help mitigate the limitations inherent to the study’s retrospective design.

## 5. Conclusions

In conclusion, our data suggest that preoperative Yttrium-90 transarterial radioembolization is a safe and oncologically advantageous approach in selected patients with hepatocellular carcinoma (HCC) who are candidates for right or extended right hepatectomy. It does not increase perioperative morbidity and may contribute to improved long-term cancer-specific and disease-free survival by promoting both tumor control and biological selection. These findings are further supported by evidence of volumetric hypertrophy, favorable recurrence patterns, and histological response, reinforcing the role of TARE as a truly active neoadjuvant treatment, particularly in cases of borderline resectability and vascular invasion.

Further prospective studies are warranted to better define its role as a neoadjuvant strategy in resectable or borderline resectable HCC, either as monotherapy or in combination with systemic treatments, within an increasingly integrated and tailored multimodal approach to HCC management.

## Figures and Tables

**Figure 1 cancers-17-02556-f001:**
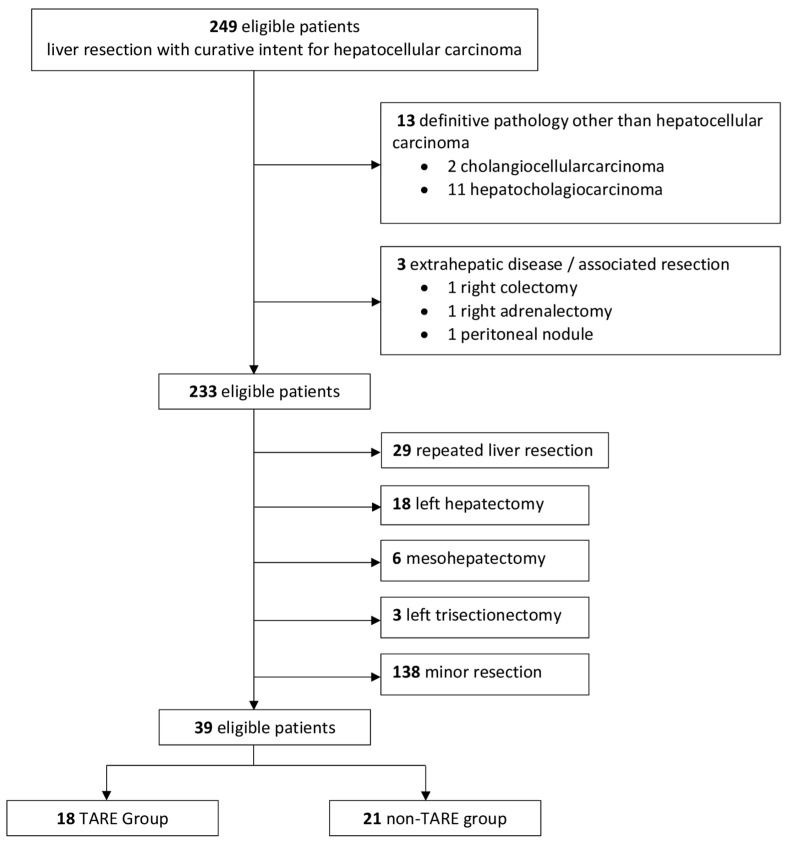
Flow diagram of patients included in the analysis. TARE: Yttrium-90 transarterial radioembolization.

**Figure 2 cancers-17-02556-f002:**
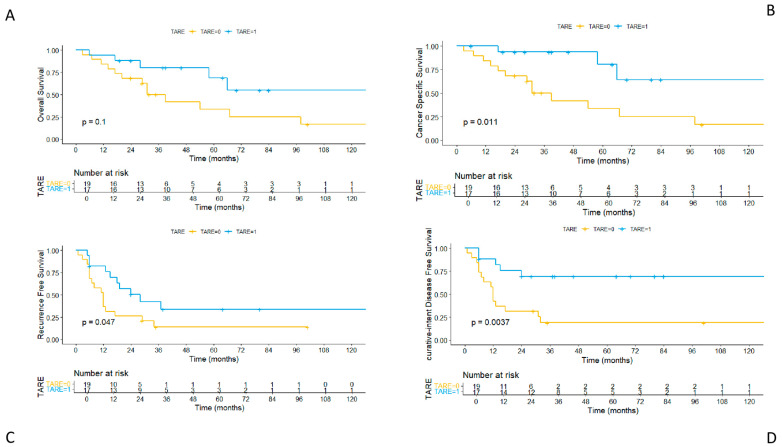
Comparison of Kaplan–Meier curves for overall survival (OS) (**A**); cancer-specific survival (CSS) (**B**); recurrence-free survival (RFS) (**C**); and curative-intent disease-free survival (ciDFS) (**D**) between the TARE group and non-TARE group.

**Table 1 cancers-17-02556-t001:** Baseline and pathological characteristics.

	Non-TARE Group *n* = 21 (53.8%)	TARE Group *n* = 18 (46.15%)	*p*-Value
Age, median [IQR]	72 [69.0, 77.0]	74.50 [70.50, 76.0]	0.489
Male (%)	16 (76.2)	17 (94.4)	0.190
BMI, median [IQR]	25.93 [21.80, 29.32]	25.96 [23.99, 28.62]	0.673
Charlson Comorbidity Index, median [IQR]	6 [5, 7]	6 [6, 7]	0.402
ASA score (%)			
1	1 (4.8)	0 (0.0)	0.537
2	6 (28.6)	3 (16.7)	
3	12 (57.1)	14 (77.8)	
4	2 (9.5)	1 (5.6)	
No Virus (%)	15 (71.4)	7 (38.9)	0.033
HBV (%)	0 (0.0)	4 (22.2)	
HCV (%)	6 (28.6)	7 (38.9)	
Metabolic Disease (%)	9 (42.9)	5 (27.8)	0.504
Child Pugh Score (%)			
Child Pugh A (%)	19 (90.5)	18 (100)	0.490
Child Pugh B (%)	2 (9.5)	0 (0.0)	
MELD, median [IQR]	7.00 [7.00, 8.00]	7.50 [7.00, 8.75]	0.588
AFP ng/mL (%)			
≤100	14 (66.7)	12 (66.7)	1.000
>100	6 (28.6)	5 (27.8)	
Bilirubin preoperative, mg/dl, median [IQR]	0.71 [0.60, 1.05]	0.77 [0.61, 0.90]	0.877
15′ ICG-RR, median [IQR]	10.30 [4.30, 12.50]	14.80 [9.85, 21.73]	0.033
Esophageal Varices (%)	1 (4.8)	2 (11.1)	0.586
Size of Node (preoperative), mm, median [IQR]	80.00 [60.00, 150.00]	70.00 [47.50, 80.00]	0.171
Single Node (%)	18 (85.7)	10 (55.6)	0.218
Multifocal Disease (%)	3 (14.3)	8 (44.4)	
Portal Vein Invasion Vp (%)			
Vp0	16 (76.2)	10 (55.6)	0.043
Vp1	0 (0.0)	1 (5.6)	
Vp2	4 (19.0)	1 (5.6)	
Vp3	1 (4.8)	6 (33.3)	
Hepatic Vein Invasion VV (%)			
VV0	10 (47.6)	12 (66.7)	0.093
VV1	0 (0.0)	2 (11.1)	
VV2	10 (47.6)	4 (22.2)	
VV3	1 (4.8)	0 (0.0)	
Right Portal Vein Invasion Vp3 (%)	1 (4.8)	6 (33.3)	0.035
Concomitant Portal and Hepatic Vein Invasion (%)	3 (14.3)	3 (16.7)	1.000
BCLC Stage (%)			0.295
A	9 (42.9)	5 (27.8)	
B	1 (4.8)	4 (22.2)	
C	11 (52.4)	9 (50.0)	
FLR/TLV pre-treatment, %, median [IQR]	40.0 [37.0–47.3]	39.1 [35.8–50.7]	0.954
FLR/TLV post-treatment, %, median [IQR]	51.9 [51.0, 53.9] *	56.9 [40.0, 62.2]	0.737

BMI: body mass index, ASA: American Society of Anesthesiologists, HBV: Hepatitis B Virus, HCV: Hepatitis C Virus, MELD: Model of end-stage liver disease, AFP: Alpha-fetoprotein, 15′ ICG-RR: indocyanine green retention rate at 15 min, BCLC: Barcelona Clinic Liver Cancer, FLR/TLV: future liver remnant/total liver volume, *: Four PVE (portal vein embolization) patients.

**Table 2 cancers-17-02556-t002:** Intraoperative outcomes.

	Non-TARE Group *n* = 21 (53.8%)	TARE Group *n* = 18 (46.15%)	*p*-Value
Operative time, minutes, median [IQR]	320 [300.00, 420.00]	330.00 [302.50, 360.00]	0.899
Operative time (liver transection), minutes, median [IQR]	90.00 [79.00, 100.00]	86.50 [67.50, 106.75]	0.888
Pringle maneuver (%)	15 (71.4)	15 (83.3)	0.464
Duration Pringle maneuver, minutes, median [IQR]	46.00 [0.00, 68.00]	53.00 [24.75, 60.50]	0.734
Blood loss, ml, median [IQR]	330.00 [190.00, 400.00]	275.00 [200.00, 570.00]	0.778
Section plane cm^2^, median [IQR]	75.00 [68.00, 101.00]	64.50 [46.00, 82.00]	0.143
Type of surgery (%)			
Right hepatectomy (%)	17 (81.0)	8 (44.4)	0.024
Extended right hepatectomy (%)	4 (19.0)	10 (55.6)	
Minimally invasive resection (%)	5 (23.8)	8 (44.4)	0.196
Intraoperative complication (%)	3 (14.3)	1 (5.6)	0.609

**Table 3 cancers-17-02556-t003:** Postoperative outcomes.

	Non-TARE Group *n* = 21 (53.8%)	TARE Group *n* = 18 (46.15%)	*p*-Value
Postoperative complications (%)	8 (38.1)	7 (38.9)	1.000
Blood transfusions (%)	3 (14.3)	1 (5.6)	0.609
PHLF (%)	4 (19.0)	2 (11.1)	0.667
Grade PHLF sec. ISGLS (%)			
Grade A	0 (0.0)	1 (5.6)	0.609
Grade B	2 (9.5)	0 (0.0)	
Grade C	2 (9.5)	1 (5.6)	
Biliary fistula (%)	3 (14.3)	2 (11.1)	1.000
Clavien–Dindo ≥3 (%)	5 (23.8)	1 (5.6)	0.190
CCI ≥ 20.9 (%)	8 (38.1)	4 (22.2)	0.322
Readmission (%)	2 (9.5)	1 (5.6)	1.000
90-day postoperative mortality (%)	2 (9.5)	1 (5.6)	1.000
Hospital stay, days, median [IQR]	8.00 [5.00, 11.00]	6.00 [5.00, 9.00]	0.318
Histology			
Cirrhosis (%)	3 (14.3)	12 (66.7)	0.001
Multinodular disease (%)	2 (9.5)	7 (38.9)	0.055
Single node disease (%)	19 (90.5)	11 (61.1)	
Satellitosis (%)	13 (61.9)	6 (33.3)	0.111
Size of node, mm, median [IQR]	80.00 [60.00, 150.00]	46.50 [36.25, 77.50]	0.020
Necrosis percentage, median [IQR]	10.00 [10.00, 20.00]	70.00 [50.00, 87.50]	0.002
Resection margin, mm, median [IQR]	10.00 [2.00, 15.00]	2.00 [1.00, 10.00]	0.032
Positive margin, mm (%)	1 (4.8)	2 (11.1)	0.586
Microvascular invasion (%)	19 (90.5)	11 (61.1)	0.055
Edmonson Grade 3–4 (%)	18 (85.7)	10 (55.6)	0.072
Tumor capsule (%)	15 (71.4)	11 (61.1)	0.520

PHLF: post-hepatectomy liver failure, ISGLS: International Study Group of Liver Surgery, CCI: Charlson Comorbidity Index.

## Data Availability

The raw data supporting the conclusions of this article will be made available by the authors on request.

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
