# Peer review of "Impact of Preoperative Yttrium-90 Transarterial Radioembolization on Patients Undergoing Right or Extended Right Hepatectomy for Hepatocellular Carcinoma"

_cancers, 2025, doi:10.3390/cancers17152556_

Round 1

Reviewer 1 Report (Previous Reviewer 2)

Comments and Suggestions for Authors

I accepted the explanation of the authors and agreed with the modification of the text of this article. I think that this version of the manuscript could be considered for acceptance for publication. 

Author Response

Reviewer 1:

Comment 1: I accepted the explanation of the authors and agreed with the modification of the text of this article. I think that this version of the manuscript could be considered for acceptance for publication. 

Response 1: We would like to express our sincere gratitude to Reviewer 1 for the positive and supportive feedback. We truly appreciate your recognition of the scientific value of our work and your acknowledgment that the revised manuscript is suitable for publication. Thank you once again for your thoughtful comments and for your constructive contribution throughout the review process.

Reviewer 2 Report (Previous Reviewer 1)

Comments and Suggestions for Authors

The paper is on a cutting-edge topic. 

Could the authors provide more details on the dose used for TARE?

How was the preoperative study of the patients conducted?

Could the authors evaluate or at least comment some baseline clinical comorbidities that could affect the patients outcomes?

Author Response

Reviewer 2:

The paper is on a cutting-edge topic. 

Comment 1: Could the authors provide more details on the dose used for TARE?

Response 1: We thank the reviewer for this relevant observation. As correctly pointed out, detailed information on dosimetry is essential to evaluate the quality and reproducibility of TARE protocols. In the original manuscript, we had already described the dosimetric approach, including both partition model and a voxel-based convolution method, aiming for a tumor-absorbed dose >120 Gy.
In the revised version, we have now added specific data regarding the actual median tumor-absorbed dose, which was 230.5 Gy [139.5–351.0], as calculated using patient-specific dosimetry. This new information has been included in the Results section, under the paragraph reporting on treatment allocation and protocol adherence.

Changes in the text 1, lines 315-316: The median tumor-absorbed dose in the TARE group was 230.5 Gy [139.5–351.0], as calculated using patient-specific voxel-based dosimetry.

Comment 2: How was the preoperative study of the patients conducted?

Response 2: We thank the reviewer for this valuable question. The preoperative evaluation protocol adopted at our institution is described in Section 2.2 “Diagnosis and preoperative staging.”
All patients underwent contrast-enhanced thoraco-abdominal CT, complete liver function testing, serum AFP measurement, and upper endoscopy. Hepatic functional reserve was assessed using the indocyanine green retention rate at 15 minutes (ICG-R15), and when necessary, liver-specific contrast-enhanced MRI with MRCP was performed.
Macrovascular invasion was classified according to standardized criteria (Liver Cancer Study Group of Japan), and all treatment decisions were made during multidisciplinary tumor board meetings, following a therapeutic hierarchy approach.
We hope this overview clarifies the rationale and structure of the preoperative work-up used in our study cohort.

Comment 3: Could the authors evaluate or at least comment some baseline clinical comorbidities that could affect the patients outcomes?

Response 3:

We thank the reviewer for this important observation. In response, we have added a comment in the Results section (3.1) to emphasize that baseline comorbidities were systematically evaluated using both the Charlson Comorbidity Index and the ASA classification system. Median Charlson scores were similar between the two groups (6 in both), and ASA distribution was not significantly different. These findings suggest a comparable comorbidity burden between the TARE and non-TARE groups, minimizing the risk of selection bias. This clarification reinforces the validity of outcome comparisons between groups.

Changes in the text 3, lines 331-332: Comorbidity burden was comparable between the two groups, as reflected by similar Charlson Comorbidity Index scores and ASA classifications

Round 2

Reviewer 2 Report (Previous Reviewer 1)

Comments and Suggestions for Authors

The manuscript is OK

This manuscript is a resubmission of an earlier submission. The following is a list of the peer review reports and author responses from that submission.

Round 1

Reviewer 1 Report

Comments and Suggestions for Authors

The article was nicely written but i have some major concerns on the retrospective design, the lack of novelty and, above all, the very limited sample size. The authors should be able to address all these points in the revision!

The authors should comment more on the role of TARE in the therapeutic management of these patients (cite the SRMA: PMID: 27579537 ). IN general, the Discussion is too short and it should be improved!

I think there were some imbalances between the two groups in terms of some baseline features. Overall, we cannot exclude the risk of selection bias.

What was the exact timing of surgery after TARE?

Reviewer 2 Report

Comments and Suggestions for Authors

Comments on TARE on hepatectomy for hepatocellular carcinoma

This article deserved great attention since it showed that Y-90 transarterial radioembolization before major surgery could be helpful for prolonging survival of patients with “borderline resectable” hepatocellular carcinoma (HCC).

Some questionable description should be re-addressed or corrected before this manuscript could be accepted for publication.

  1. I cannot agree with the statement “…approximately 70% are diagnosed at an advanced stage”. The authors cited from the reference 1, however, the commonly noted staging of HCC is not like this kind of distribution.
  2. The authors said that in the whole 39 cohort, there were 23 patients that were found to have macrovascular invasion. However, in the Table 1, the number of patients with portal vein invasion (from Vp1 to VP3) and hepatic vein invasion (from VV1 to VV3) is 30, not consistent with that described in the section of result. From Table 1, the number of patients with concomitant vascular involvement cannot be counted.
  3. These patients did not show how is their status by the most commonly used BCLC classification .
  4. The resectability of hepatic lobes is usually judged by ICG retention rate and Child-Pugh classification of liver function reserve. Therefore, not out of expectation, these patients can undergo operations. However, in the BCLC classification of HCC, the common consensus is surgical intervention will not be suitable for class C patients, that is, with vascular invasion. Can the results of this manuscript overturn the consensus for management for HCC patients beyond class C?